# Metastatic adrenocortical carcinoma displays higher mutation rate and tumor heterogeneity than primary tumors

Sudheer Kumar Gara[1], Justin Lack[2], Lisa Zhang[1], Emerson Harris[1], Margaret Cam[2] & Electron Kebebew[1,3]

Adrenocortical cancer (ACC) is a rare cancer with poor prognosis and high mortality due to metastatic disease. All reported genetic alterations have been in primary ACC, and it is unknown if there is molecular heterogeneity in ACC. Here, we report the genetic changes associated with metastatic ACC compared to primary ACCs and tumor heterogeneity. We performed whole-exome sequencing of 33 metastatic tumors. The overall mutation rate (per megabase) in metastatic tumors was 2.8-fold higher than primary ACC tumor samples. We found tumor heterogeneity among different metastatic sites in ACC and discovered recurrent mutations in several novel genes. We observed 37–57% overlap in genes that are mutated among different metastatic sites within the same patient. We also identified new therapeutic targets in recurrent and metastatic ACC not previously described in primary ACCs.

[1] Endocrine Oncology Branch, National Cancer Institute, National Institutes of Health, Bethesda, MD 20892, USA. [2] Center for Cancer Research, Collaborative Bioinformatics Resource, National Cancer Institute, National Institutes of Health, Bethesda, MD 20892, USA. [3] Department of Surgery and Stanford Cancer Institute, Stanford University, Stanford, CA 94305, USA. Correspondence and requests for materials should be addressed to E.K. (email: kebebew@stanford.edu)

Adrenocortical carcinoma (ACC) is a rare malignancy with 0.7–2 cases per million per year[1,2]. The five-year survival rate for patients with resectable tumors is less than 30%. Particularly, patients with metastatic disease have a median survival of less than 1 year. Unfortunately, most patients with ACC have locally advanced cancer or metastasis at the time of diagnosis. Deaths due to ACC are associated with metastatic disease.

Our understanding of the pathogenesis of ACC has been greatly improved over the past decade. Molecular studies have demonstrated that *TP53* inactivating mutations, *CTNNB1* activating mutations, IGF2 overexpression, damaging mutations in *ZNRF3*, and high-level amplification of *TERT* are common and key drivers of ACC[3–6]. Furthermore, the international consortium of The Cancer Genome Atlas (TCGA) has identified additional ACC driver genes including *PRKAR1A*, *RPL22*, *TERF2*, *CCNE1*, and *NF1*[7]. This study further showed that a whole-genome doubling event is a marker for ACC progression and prognosis[7]. Multiple studies have reported that activating and inactivating alterations in the TP53/Rb and WNT pathways are key molecular events in the pathogenesis of ACC[4,7]. However, all of the above studies are confined to primary ACC, and death from ACC is primarily due to recurrent or metastatic disease.

Despite recent advances in our understanding of ACC, the therapeutic options for ACC are still limited, and treatment with combination mitotane-etoposide, doxorubicin, and cisplatin results in low response rates[8,9]. Studies on targeted therapeutic options using IGF-targeting agents or tyrosine kinase inhibitors have not shown good efficacy[10–12]. It is evident that most of the advanced or late-stage cancers or treatment-resistant tumors adapt differently and gain additional mutations. Mutations analysis of late-stage cancer tumor sites and treatment-resistant tumors has shown biologically informative genomic alterations. Given this, it is important to understand the nature of recurrent and metastatic ACC to inform candidate genetic changes involved in cancer progression and to identify therapeutic targets. Therefore, our study objectives were to analyze the genomic landscape of metastatic ACC, the nature of different metastatic tumor sites (tumor heterogeneity), and their commonality and differences compared to primary ACC.

## Results

**Metastatic ACC has a higher mutation rate.** We performed whole-exome sequencing on 33 histologically confirmed metastatic ACCs (lung, liver, pancreas, kidney, peritoneum, and other tissue sites with matched peripheral blood sample DNA) collected from 14 patients with metastatic ACC. We identified 15,321 somatic mutations in the coding region of the genome in 33 tumors; 5928 nonsynonymous mutations and 9393 silent mutations (Supplementary Table 1). The mean somatic mutation rate in the coding region of metastatic ACC was 10.17 mutations per megabase, with 3.93 and 6.24 mutations per megabase for nonsynonymous and silent mutations, respectively, (Supplementary Table 1). We compared the overall somatic mutation burden between our cohort of metastatic ACC and the primary ACC tumors from the TCGA, reprocessed through our somatic variant calling pipeline (see methods) to eliminate pipeline-driven difference in variant call rates. The ACC metastatic tumors had 2.8-fold higher median mutation rate compared to primary ACC (Fig. 1a). Since the mutation rate was compared between primary ACC from the TCGA cohort to the metastatic cohort, we performed the same analysis in one matched primary ACC with a metastatic lung ACC tumor from the same patient. We found that metastatic lung ACC had a threefold higher mutation rate compared to the primary ACC (Fig. 1b). In addition, we compared the mutation rate of metastatic ACC with other cancer types including primary ACC from the TCGA to understand our findings in the context of not only primary ACC but also other primary cancer types (Fig. 1c). Next, we tested the idea whether primary ACC tumors of the TCGA from patients with metastasis has any effect on mutation burden and found no significant difference in the mutation rate by tumor stage supporting our findings that the mutation burden is higher in metastatic ACC as compared to primary ACC (Fig. 1d). We also analyzed the relative proportions of the six possible base-pair substitutions in the transcribed and non-transcribed strands among all the metastatic ACC tumors to understand their relevance with other solid tumors including primary ACC tumors. Similar to primary ACC from the TCGA data and most other solid tumors[13,14], metastatic ACC were characterized by a predominance of C > T transitions (Fig. 1e). However, we did not notice any significant difference of T > C mutations between transcribed and non-transcribed strands in metastatic ACC, unlike in primary ACC tumors (Fig. 1e, f).

We next analyzed genes recurrently mutated in multiple metastatic ACC tumor samples and compared their status in primary ACC from the TCGA (Fig. 1g). Similar to primary ACC, we observed that *CTNNB1*, *DNHD1*, and *TTN* were frequently mutated in metastatic ACC. Nevertheless, we have also identified genes such as *ENTHD1*, *HELZ2*, *PCDH12*, *SHANK1*, and *WDR66* that were more frequently mutated in metastatic ACC but not in primary tumors (Fig. 1g). Next, we selected the frequently mutated genes in primary ACC and compared their mutation status in metastatic ACC (Fig. 1h). As expected, the majority of the genes that are mutated in primary ACC were also mutated in metastatic ACC (Fig. 1h). We also validated 5/5 tested mutations using droplet digital polymerase chain reaction (ddPCR) and 72 of 77 mutations using Sanger sequencing (Supplementary Figs. 1, 2). In addition, we noticed that two metastatic ACCs from the same patient (Case 1) had a strong hypermutation phenotype (Supplementary Table 1), and two additional patients had > /10 single-nucleotide polymorphisms (SNPs) per megabase compared to a median of 2.63 SNPs per megabase for all metastatic ACCs in our data set. To determine if these hypermutation phenotypes are driven by DNA repair defects, we examined copy number variations (CNVs), somatic SNPs and insertions and deletions (INDELs), and germline SNPs and INDELs for putative loss-of-function (LoF) mutations (homozygous deletions; nonsense, non-frameshift INDELs > = 3 amino acids; and frameshift mutations in the Wood DNA repair genes)[15]. For the two tumor samples with the highest mutation rates, the only LoF mutation shared between both tumors is a heterozygous 9-base germline insertion in exon 1 of *MSH3* (Supplementary Table 2 and Supplementary Fig. 3), which was amplified in copy number and had (allele frequency > 0.9 in both samples) complete loss of heterozygosity (LOH) in both tumor samples. Somatic INDELs overlapping this mutation were also detected for three primary ACCs, and while it is unclear if any of these samples have LOH and fixation of these alterations, one of the three samples (TCGA-PK-A5HB-01) has a hypermutation phenotype, possessing the second-highest mutation rate of the 92 primary ACCs available (24). For the ACC metastasis with the second-highest mutation rate (Tumor 20; Supplementary Table 1), no germline LoF events were detected, but multiple somatic LoF events were detected that may be contributing to its hypermutation phenotype. We detected a focal homozygous deletion of *MSH6* (Supplementary Table 2 and Supplementary Fig. 4), which was not detected in any of the 92 primary ACC tumors available in cBioPortal (Supplementary Table 2 and Supplementary Fig. 3). In addition, we detected a frameshift mutation in *ATM* that has become homozygous through an LoH event (Supplementary Table 2 and Supplementary Fig. 4B) and is fixed in the tumor sample (allele frequency =

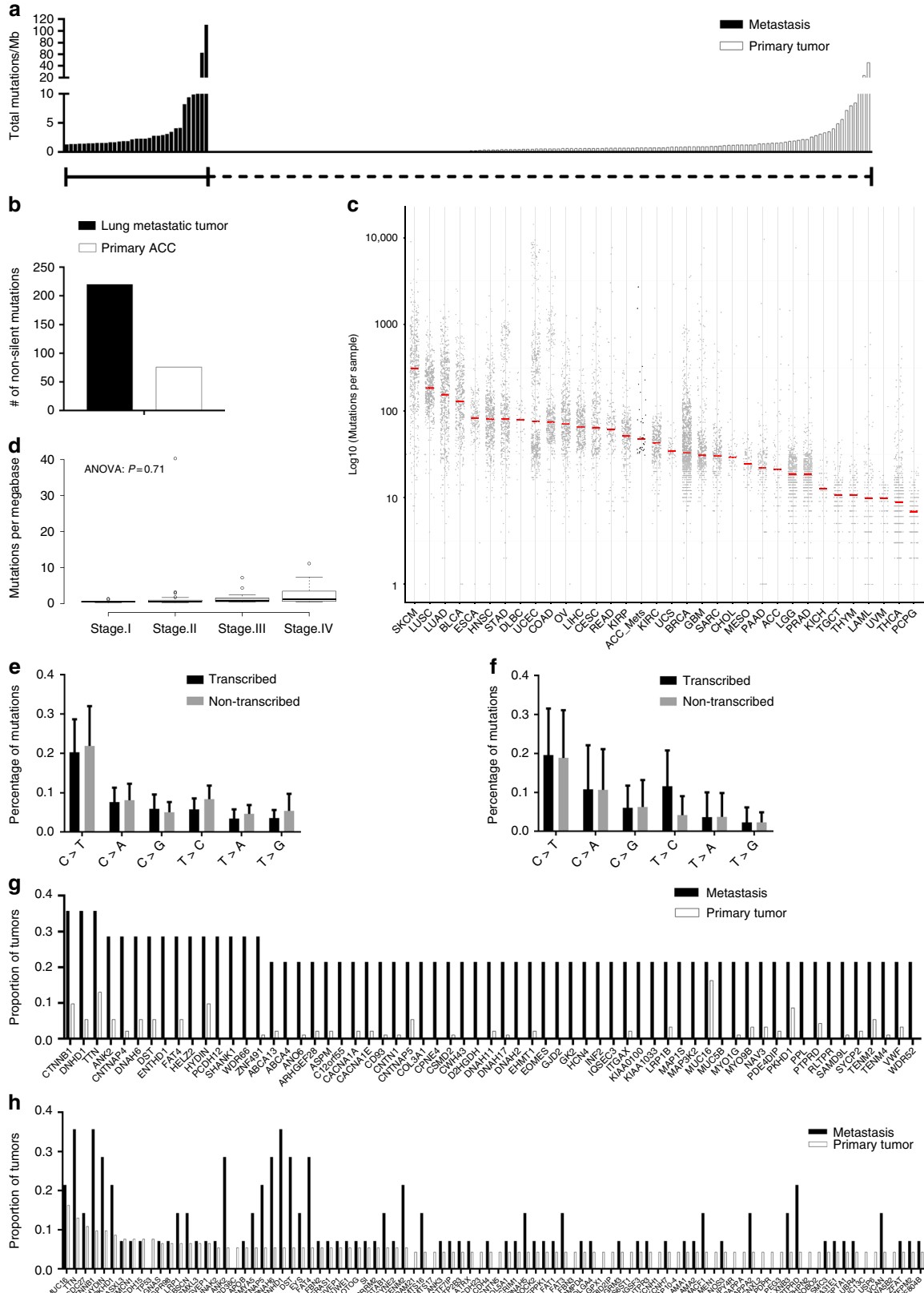

**Fig. 1** Metastatic ACC has a higher mutation burden compared to primary ACC. Total number of mutations per megabase in each tumor sample of both metastatic and primary ACC (TCGA). Each bar on *x*-axis represents a patient or a tumor sample (**a**). Total mutations per megabase in metastatic lung ACC and its counterpart primary ACC tumor from the same patient (**b**). Metastatic ACC mutation rate compared to other cancer types including primary ACC (TCGA) (**c**). Mutation rate in the primary ACC (TCGA) based on tumor stage (**d**). Relative proportions of the six possible base-pair mutations in 33 metastatic ACC (**e**) and primary ACC (**f**). The top genes that are frequently mutated in metastatic (**g**) and primary ACC (**h**)

0.955). Interestingly, a single sample from the TCGA ACC primary tumors (TCGA-OR-A5K4-01) also possessed LoF mutations in *MSH6* and *ATM* (although both were nonsense mutations, Supplementary Fig. 5), and had the eighth-highest mutation rate of that sample of 92 primary tumors (24), suggesting compound LoF in these two genes may be driving hypermutation. The final patient with hypermutating ACC metastases had no detected LoF mutations in any DNA repair genes that were present in all three of the metastases sampled for that patient. However, there were two rare germline *ATM* missense mutations that rose to fixation in all three of the tumor samples from that patient through complete LoH (Supplementary Table 2 and Supplementary Fig. 6).

**Metastatic ACCs are highly heterogeneous**. To understand the nature of metastatic ACC, we selected all the somatic mutations within the coding regions of each tumor sample site and performed principal component analysis (PCA) on the entire patient cohort. We found that metastatic ACC tumor samples clustered by patients and not by location of the tumor metastases (Fig. 2a), as is expected given their origin from the same primary tumor. Next, we analyzed each metastatic tumor site to identify the genes that are frequently mutated in each tumor site. Since one of the patient (Case 1) had a hypermutation phenotype in both the metastatic sites (lung and other tissue site), we have performed the heatmaps and phylogenies to identify the overlapping and non-overlapping variants with and without this patient (Supplementary Fig. 7 and Fig. 2b, c). Consistent with PCA, the majority of mutated genes were not common across metastatic tumor sites (Fig. 2b–e). Nevertheless, we observed that genes such as *DNAH6*, *MUC5B*, *HELZ2*, and *KIAA0100* were frequently mutated in three of six lung ACC metastases (Supplementary Table 3). In addition, *CTNNB1* was mutated in both ACC liver metastases (Fig. 2d). Genes such as *ARHGEF28* and *PPL* were mutated in three of six ACC samples classified as other tumor sites (Fig. 2c and Supplementary Table 4). We did not see any common mutations in metastatic peritoneal tumor sites (Fig. 2e).

**Metastatic ACC tumor sites within a patient are homogenous**. As we did not find significant commonality among metastatic tumor sites across patients, we decided to analyze the status of the different metastatic ACCs within a given patient. We selected four patients with metastases to more than one tumor site and found that most mutated genes were common within a patient regardless of the site of metastases (Fig. 3a–d). For example, we observed that 57%, 44%, and 37% of the genes that were mutated in Case 1, Case 2, and Case 3, respectively, were common regardless sites of tumor metastases (Fig. 3a–c). Particularly, among the 16 mutations that are shared between the three tissues, genes such as *CTNNB1*, *IGF2R*, and *SF1* that were known to play an important role in the pathogenesis of adrenocortical tumor were present (Fig. 3a). Higher number of additional shared mutations in genes (five) between kidney and liver are present when compared to only one mutation in gene between kidney and peritoneum (Fig. 3a). Although there are 20 shared mutations in genes between lung and other tissue site, *CTNNB1* is the only known common gene that is mutated between these tumor tissues (Fig. 3b). However, we noticed that only 14% of the genes mutated in the lung metastases and other tumor sites were common in Case 4, possibly because this patient had a hypermutation phenotype (Fig. 3d).

**Copy number variation in metastatic ACC tumors**. At the genome-wide scale, copy number variation in metastatic tumors (Fig. 4) appears to be very similar to that of primary tumors (see

refs. [4,7] for primary tumor CNV patterns) and is essentially identical to that of Assie et al.[4]. In general, large-scale LoH is typical of primary ACC[4,16] and was readily apparent in our metastatic samples, with an average of 49.6% LoH for our samples (Fig. 4, Supplementary Table 1). However, in contrast to primary tumors[16], we detected no whole-genome doubling (WGD) events in the ABSOLUTE analysis of our ACC metastases, in spite of the fact that the proportion of CNV-altered genomes ranged from 0.17 to 0.96 (Supplementary Table 1). Zheng et al.[7] pointed out that these "noisy" CNV hypervariable samples consistently had WGD events, but we failed to detect this signal in our metastatic ACC samples. Furthermore, clonal diversity was extremely low, with the subclonal genome fraction ≤ 0.02 across all samples. At the gene level, the *TERT* amplification, *CHEK2/ZNRF3* deletion, and *CDKN2A* deletion previously identified as being common in primary ACC were also identified in our metastatic sample. However, none of our samples were positive for deletions in *RB1*, 3q13.31, or 4q34.3, all of which were identified as recurrent in primary ACC[4,16].

We next examined variation among metastatic sites in terms of homozygous deletions and high-level amplifications (five or more copies). Only one homozygous deletion on chromosome 9 (*CDKN2A*) is common in metastatic lung and other tissue (three out of six patients) when compared across different patients (Supplementary Fig. 8A, B and Supplementary Fig. 9). On the other hand, we did not observe any common high-level amplifications in both metastatic lung and other tumor tissues across different patients (Supplementary Fig. 8C, D). However, we found common homozygous deletions in *CDK11A/B*, *STK4/TOMM34*, and *CDKN2A/MTAP* (Supplementary Figs. 10, 11), and in lung and other tissue of the same patient. Meanwhile, common amplifications were observed in 15 genes in two different metastatic tumors of the same patient (Supplementary Fig. 8E, F). In agreement with the patterns observed for SNPs, copy number variation in metastatic tumors was also highly homogenous within patients but divergent between metastases in the same site (or different sites) from different patients.

**Molecular pathways associated with metastatic ACC**. We analyzed the entire gene list of recurrent mutations in metastatic ACC through ingenuity pathway analysis (IPA) to identify molecular pathways associated with these tumors. Summarized in Fig. 5a are the key networks of the altered pathways in these tumors. Some of the top altered pathways—with the total number of overlapping genes in our cohort and the total number of genes associated in that pathway are also summarized in Table 1. In particular, four signaling pathways, including ERBB4, retinoic acid receptor (RAR), G-protein-coupled receptor (GPCR), and platelet-derived growth factor receptor (PDGFR), were frequently altered in metastatic ACC (Fig. 5b–d).

**Novel drug candidates for metastatic ACC**. We analyzed targetable recurrent mutations in ACC metastases using commercially available and FDA-approved drugs through the IPA database. The majority of the drug targets (52%) were membrane proteins comprising transmembrane, GPCRs, transporters, and ion channels. Twenty-two percent of the drug targets were cytoplasmic, including kinases, phosphatases, and enzymes, whereas 11% of them were nuclear (Fig. 6a, b). About 15% of the drug targets were extracellular, mainly cytokines and growth factors. We selected all the drugs that can be used to treat multiple patients with metastasis in our cohort based on the mutational spectrum (Fig. 6c). We identified drugs such as Afatinib, Cabozantinib, and Sunitinib that could target receptor tyrosine kinases ERBB4, AXL, and FLT1/3, respectively. Drugs targeting

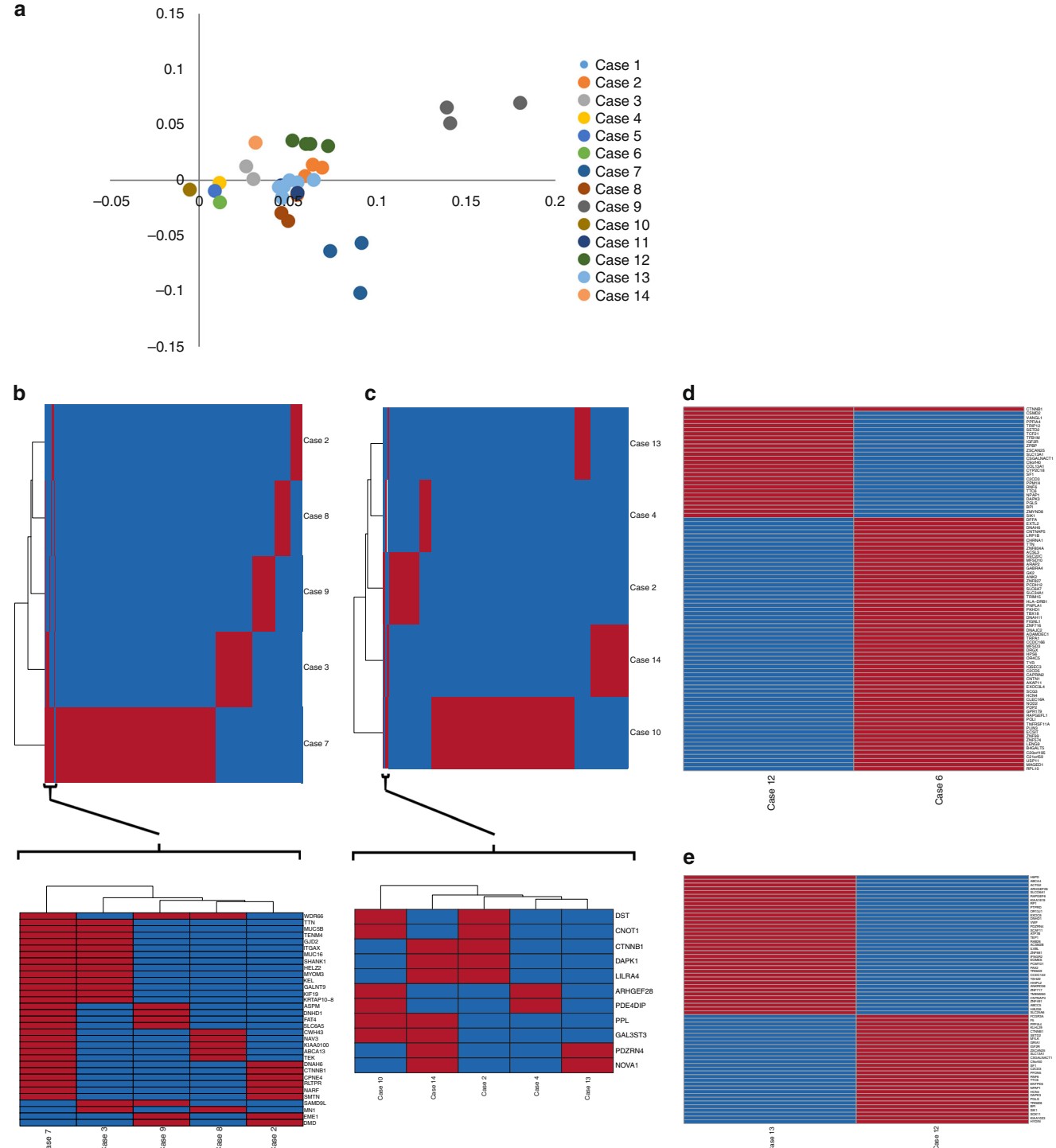

**Fig. 2** Genotyping of metastatic ACC shows tumor heterogeneity among different metastatic tumor sites. The principal component analysis (PCA) of all the missense, nonsense, and splice site mutations in 33 metastatic adrenocortical tumors (**a**). Heatmap and clustering of the overlapping variants within the genes in metastatic lung (**b**) and other tumor sites (**c**) from five different patients (Case 1 with hypermutation phenotype was excluded). The region containing maximum similarity was reanalyzed to better represent the shared genes that are mutated. Heatmap of the total number of overlapping and non-overlapping variants within the genes that are mutated in multiple patients with metastasis in liver (**d**), and peritoneum (**e**). Red bar in the heatmap indicates the gene is mutated, whereas the blue represents that the corresponding gene is wild-type in each sample

GPCRs and ion channels were also among the list that can be explored in ACC. Collectively, based on whole-exome sequencing mutations identified in metastatic ACC, 9 of 14 patients could have treatment using commercially available and approved drugs that target alterations in their tumors.

## Discussion

In this study, we provided data on the types of mutations present in metastatic ACC and the higher mutation rate in metastatic ACC relative to primary ACC. In addition, we demonstrate that metastatic ACCs are highly heterogenous across patients but also

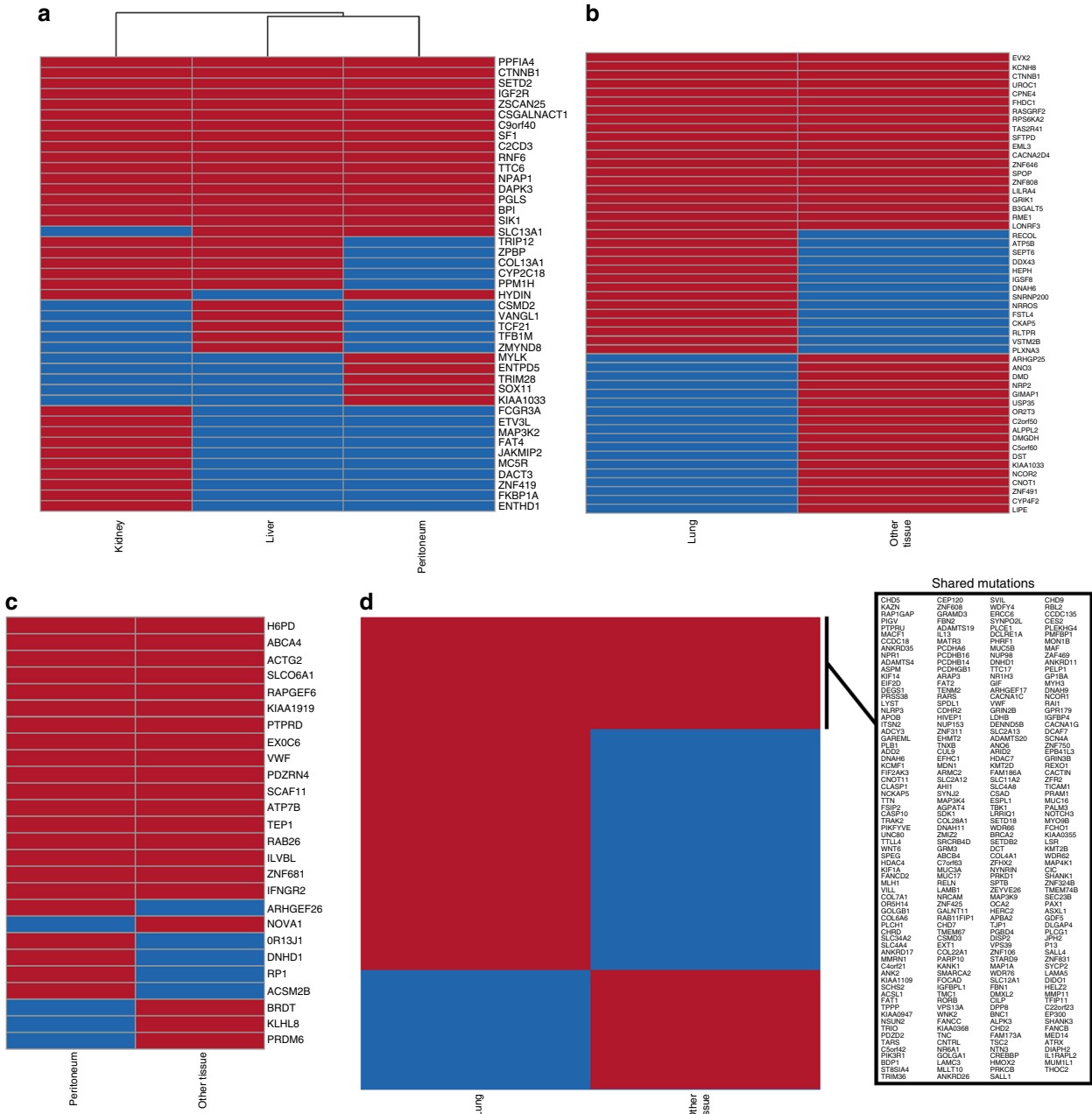

**Fig. 3** Different metastatic ACC sites within the same patient are predominantly homogenous. Heatmap and clustering of the total number of overlapping and non-overlapping variants within the genes that are mutated in multiple tumor sites from each patient (**a–d**). The list of genes with shared mutations between lung and other tissue in the hypermutated patient (**d**) were shown in a box. Red bar in the heatmap indicates the gene is mutated, whereas the blue represents that the corresponding gene is wild-type in each sample

share many similarities between different metastatic tumors within the same patient. Furthermore, we identified novel pathways that have not been reported in association with ACC, such as ERBB4, retinoic acid receptor, GPCR, and PDGFR signaling, that are genetically altered and are potentially targetable with currently available drugs.

Although it is still unclear, some studies suggest that ACC is a multistep process in which several genome-wide alterations are accumulated over time[17,18]. Moreover, it is evident that metastatic tumors often undergo genomic evolution during progression and drug resistance, thereby accumulating additional mutations. Therefore, the higher mutation rate we observed in metastatic ACC is not surprising and has been observed in metastatic tumors from prostate, ovarian, colorectal, and breast

cancers as compared to their corresponding primary tumors[19–22]. For the same reason, one would expect the candidate driver genes of metastasis will be different compared to primary tumors. Therefore, understanding candidate driver genes in primary tumors alone may not suffice when treating patients with metastatic disease.

Multiple genes such as *CSMD2* (CUB and Sushi Multiple Domains 2), *LRP1b* (LDL Receptor related Protein 1B) and *KIAA0100*, which have been suggested to have tumor suppressor function in the literature were also found to be mutated in metastatic ACC tumors[16,23–25]. Particularly, *CSMD2*, a candidate tumor suppressor gene in colorectal cancer and breast cancer patients was mutated in three metastatic ACC samples (Fig. 1g). *LRP1b*, a candidate tumor suppressor gene that is frequently

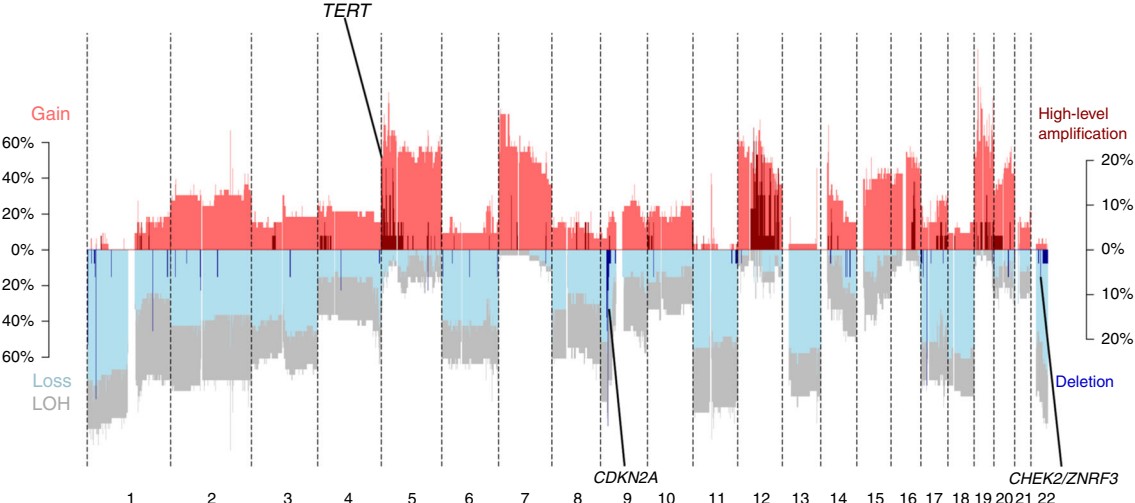

**Fig. 4** Copy number variation in metastatic tumors. High-level amplifications and deletions in metastatic ACC tumors. Regions of *TERT*, *CDKN2A*, and *CHEK2/ZNF3* that are common in primary tumors and observed in metastatic tumors are highlighted. Red and blue represent chromosomal gain and loss, respectively

inactivated in non-small lung cancer cells was also mutated in three metastatic ACC samples (Fig. 1g). Finally, *KIAA0100*, a candidate tumor suppressor gene in acute monocytic leukemia was found to be mutated in three metastatic ACC samples (Fig. 1d). However, these mutations in metastatic ACC were missense mutations. We did not find any mutations in known or suspected oncogenes in the literature to be frequently mutated in metastatic ACC. Primary ACC are enriched for mutations in *CTNNB1*, *TP53*, *ZNRF3*, *DAXX*, and *MEN1* genes, including homozygous deletions or high-level amplifications[4,7]. Although we observed a higher frequency of mutations in the *CTNNB1* (p. D32G, p.G34R, p.S45P, and p.S45F) in metastatic ACC, there were no mutations observed in *TP53*, *ZNRF3, MEN1*, and *DAXX*. However, it should be noted that only 7 of 91, 4 of 91, and 2 of 91 primary ACCs in the TCGA study were found to have damaging mutations in *TP53*, *MEN1*, and *DAXX*, respectively. It appears that metastatic ACCs do adapt and gain mutations in other genes that are different than primary ACCs. The majority of the recurrent mutations in the metastatic ACC samples were missense mutations. Although, we found frameshift-, nonsense- or splice site mutations in metastatic ACC tumors, the majority of them were not recurrent (present in multiple metastatic samples) and are, therefore, not discussed here. Many of the genes that are frequently mutated in metastatic ACCs alone are known to be associated with other types of cancers. For instance, somatic mutations in *CSMD2* was reported in non-small cell lung cancers[26]. Particularly, loss of *CSMD3* has been shown to increase the proliferation of airway epithelial cells[26]. The extracellular matrix gene, *COL3A1*, which was originally discovered to cause autosomal-dominant Ehlers-Danlos syndrome, was also recently found to be significantly altered in patients with melanoma[27]. Homozygous LoF mutations in *RLTPR* (*CARMIL2*) has been associated with disseminated EBV + smooth muscle tumors[28]. In contrast, some of the genes, including *ENTHD1*, *HELZ2*, *PCDH12*, *CPNE4*, and *SHANK1*, that are mutated in metastatic ACCs alone are completely novel and have not been functionally characterized in any cancer type until now. However, some of these variants are present in the TCGA data set of other cancer types. For example, the *P1772S* variant in the *HELZ2* gene is present in cutaneous melanoma and colorectal adenocarcinoma, the *C1183S* variant in *PCDH12* is present in renal cell carcinoma and colorectal adenocarcinoma. Somatic mutations in *ENTHD1* has also been identified in multiple breast cancer cases[29]. A recent

report suggested *WDR66* as an oncogene and a marker for risk stratification in esophageal squamous cell carcinoma[30]. It is tempting to speculate that some of the frequently mutated genes in primary ACCs, including *TP53*, *ANKRD36C*, *ADAM21*, and *CDH23*, were not mutated in metastatic ACC, suggesting that these tumors have lower metastatic potential. Nevertheless, our results provide evidence that metastatic ACCs do evolve and gain mutations that are different than primary ACCs. This makes it challenging for treatments, as most of the targeted therapeutic options are based on the mutational status of the primary ACCs and may explain the lack of response to treatment observed in ACC clinical trials. Although we do not provide any evidence that chemotherapy resistance driver genes are common in metastatic tumors and moreover it is very difficult to make a connection between heterogeneity and primary resistance, it is tempting to speculate high heterogeneity in tumors could lead to rapid acquired resistance because of the higher probability of pre-existing drug-resistant subclones, which may explain resistance to systemic chemotherapy that are common in ACC.

Multiple studies including TCGA have revealed key molecular pathways in different cancer types based on whole-exome sequencing, copy number and/or genome-wide gene expression data[31–36]. Molecular characterization of primary ACCs suggests that alterations in Wnt/beta-catenin and TP53/Rb signaling and chromatin remodeling are primary events[4,37]. Although many metastatic ACC tumor samples harbored mutations in the beta-catenin gene like primary ACC, other genes involving molecular pathways such as ERBB4, GPCR, RAR, and PDGFR signaling were also frequently mutated in metastatic ACC. Activating mutations in *ERBB4* have recently been reported in non-small cell lung cancer[38]. In melanoma, *ERBB4* mutations have been shown to be the major oncogenic driver with both aberrant ERBB4 and PI3K-AKT signaling[39]. Likewise, we observed mutations in ERBB4 and PI3K-AKT signaling genes in metastatic ACC. Conversely, RAR signaling is dysregulated in many cancers, but mutations in this receptor family are not a commonly observed phenomenon[40–42]. Activating mutations of G-protein-coupled receptor are mutated in approximately 20% of all cancers including skin, prostate, breast, thyroid, liver, kidney, pancreas, skin, ovary, and large intestine[43]. Particularly, the glutamate family of G-protein-linked receptors (GRM1-8) that were seen in our cohort was mutated in 8% of non-small cell lung cancers[43]. This family of proteins was also frequently altered in metastatic

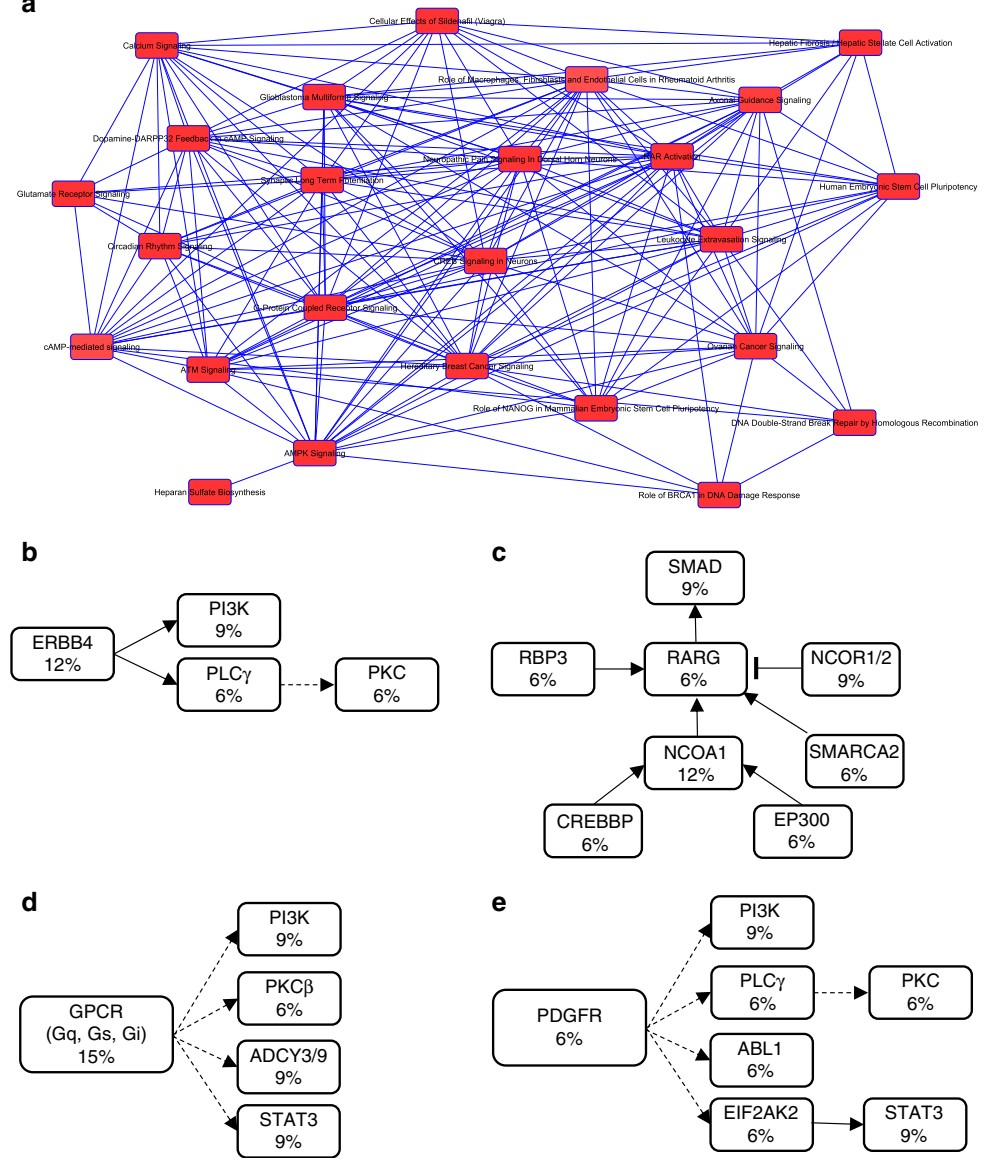

**Fig. 5** Key molecular pathways altered in metastatic ACC. Network of canonical pathways that are significantly altered in metastatic ACC based on ingenuity pathway analysis (IPA) (**a**). Pathways that are predominantly altered in metastatic ACC including ERBB4 signaling (**b**), retinoic acid receptor signaling (**c**), G-protein-coupled receptor signaling (**d**), and platelet-derived growth factor receptor signaling (**e**). The number within each box represents the percentage of gene mutations in these tumors

tumors arising from melanoma, prostate, large intestine, and lung cancers[44].

As mentioned before, targeted therapeutics for ACC are still under investigation. We found the druggable target *ERBB4* to be frequently mutated in metastatic ACC and we propose exploring the possibility of targeting this class of molecules. For instance, lapatinib is already under clinical trial in patients with stage IV melanoma with *ERBB4* mutations. However, it has not been tested in patients with advanced metastatic ACC. On the other hand, sunitinib has been tested in a phase II study in patients with ACC and is reported to have no response but prolonged tumor stabilization in 62% of the assessed patients[45]. Another case report of metastatic ACC with aberrant vascular endothelial growth factor (VEGF) expression showed significant antitumor activity following sunitinib treatment[46]. It appears that sunitinib interaction with mitotane drastically reduces its antitumor activity, implying that prior treatments of the patients are very

critical in a complete assessment of the drug potential in addition to the mutation status of metastatic ACC site(s)[47].

Most importantly, our study raises the question whether precision medicine is a better approach to treat patients with metastatic ACC. With rapidly advancing technologies and decreasing costs of genome profiling, precision medicine could certainly be one of the most cost-effective and therapeutically successful options in the control of aggressive ACC based on the tumor mutation status.

## Methods

**Tissue samples**. Metastatic adrenocortical tissue and blood samples were collected on an institutional review board-approved clinical protocol (NCT01005654 and NCT01348698, National Cancer Institute at the National Institutes of Health). Written informed consent was obtained from the patients. We included 33 human metastatic adrenocortical tissues collected from lung, liver, kidney, pancreas, peritoneum, and other tissues (retroperitoneal perinephric/perisplenic/peripancreatic/para-aortic tissue or metastatic nodule on gallbladder) for 14 patients with

metastatic disease. All diagnosis was confirmed by an endocrine pathologist based on the Weiss criteria, and tumor samples were confirmed to contain ≥ 90% tumor cells/nuclei. Tumor samples were obtained at surgical resection and immediately snap frozen and stored at −80 °C.

**Table 1 Key molecular pathways and the total number of overlapping genes in each pathway in metastatic ACC tumors**

| Pathway | Number of overlapping genes | Total number of genes |
|---|---|---|
| Hepatic stellate cell activation | 24 | 183 |
| ERBB4 signaling | 7 | 72 |
| AMPK signaling | 19 | 189 |
| RAR activation | 18 | 190 |
| GPCR signaling | 22 | 270 |
| DNA double strand break signaling | 4 | 14 |
| Cancer Metastasis signaling | 19 | 247 |
| PDGF signaling | 9 | 90 |
| PPAR signaling | 9 | 93 |

**DNA extraction**. Total genomic DNA was extracted from freshly frozen tumor tissues using a prep kit from Qiagen. The germline DNA was extracted from blood using a Qiagen Blood DNA Prep kit. All the procedures were done based on manufacturer's recommendations.

**Whole-exome sequencing and variant calling**. Tumor and germline blood DNA was used for whole-exome sequencing using the Agilent SureSelect v5 all exon + UTR (Agilent Technologies UK). Ten micrograms of genomic DNA were isolated from blood samples, and 125 base-pair paired-end reads were generated on the Illumina HiSeq 2000 (Illumina, Inc., San Diego, CA). All NGS data processing was done using our in-house developed pipeline [https://github.com/CCBR/Pipeliner)]. To mitigate the impact of pipeline-specific differences in somatic variant call rates, all of the TCGA primary ACC BAM files were downloaded from the Genomic Data Commons (GDC) data portal and passed the exact same pipeline as our metastatic tumor samples. Short read data was trimmed for the presence of adaptors and low quality using Trimmomatic v0.36[48] and the following parameter settings: Leading:10; Trailing:10; Sliding window:4:20; Minlen:20). Reads were then mapped to the hs37d5 (with decoys; ftp://ftp.1000genomes.ebi.ac.uk/vol1/ftp/technical/reference/phase2_reference_assembly_sequence/hs37d5.fa.gz) reference genome using BWA-mem v0.7.15 with default parameter settings (https://arxiv.org/abs/1303.3997). The resulting BAM files were sorted using SAMtools v1.317 and PCR duplicates were marked using Picard v2.1.1 (https://broadinstitute.github.io/picard/). Realignment around INDELs and base recalibration was performed using the Genome Analysis Toolkit v.3.4 (GATK, Broad Institute, Cambridge, MA), following the GATK Best Practices[49,50]. For somatic SNP detection, we used MuTect[51] with paired tumor/normal and run in high confidence mode. For somatic INDEL detection, we used MuTect2 [https://software.broadinstitute.org/gatk/documentation/tooldocs/current/org_broadinstitute_gatk_tools_walkers_cancer_m2_MuTect2.php)]. For germline SNP and small INDEL calling, we used the HaplotypeCaller from the GATK package[50]. For copy number analysis, we used PSCBS segmentation[52] implemented

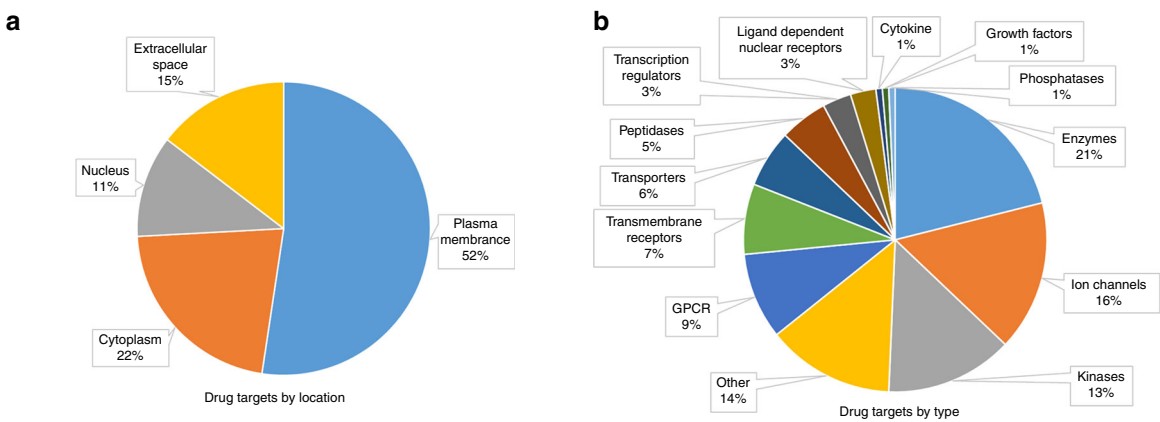

c

| Drug | # of patients | Target class | Target candidates |
|---|---|---|---|
| BMS-599626/Afatinib | 4 | Kinase | ERBB4 |
| Bumetanide | 3 | Transporter | SLC12A5, SLC12A2 |
| Cabozantinib | 3 | Kinase | NTRK2, AXL, TEK |
| Dalfampridine | 3 | Ion channel | KCNA1, KCNA10, KCNA3, KCNB2, KCNC2, KCNC3, KCND2 |
| Icosapent | 3 | GPCR and transporter | FFAR1, SLC8A1 |
| Amlodipine | 2 | Ion channel | CACNB2, CACNA2D1 |
| Carbidopa/levodopa | 2 | GPCR | DRD1, DRD5 |
| Fasorectam | 2 | GPCR | GRM3. GRM6, GRM8 |
| Nicorandil | 2 | Ion channel | KCNJ2, KCNJ12 |
| Paliperidone | 2 | GPCR | ADRA1A, ADRA1D, ADRA2B |
| Pregabalin | 2 | Ion channel | CACNA2D3, CACNA2D4 |
| Sunitnib | 2 | Kinase | FLT1, FLT3, RET |

**Fig. 6** Potential drug candidates for metastatic ACC. Recurrent gene mutations in metastatic ACC and the druggable targets based on location (**a**) and type (**b**). List of selected drugs and number of patients with mutations that can be targeted with the agent (**c**)

in a CNVkit v0.8.5[53]. ABSOLUTE v1.4[54] was used to estimate tumor purity and assess the subclonal genome fraction for each tumor.

The transcribed and non-transcribed regions were defined based on widely used annotation tool (Ensembl v91) of human genome and transcriptome. Since the libraries are stranded from the antisense strand, all the library effectively comes from (-) strand by which we calculated the mutations of genes on each strand.

**Droplet digital PCR.** Each 20 μl PCR reaction consisted of 10 μl of 2x ddPCR Supermix for probes (no dUTP) and 1 μl of 20x mutant target (FAM) and wild-type (HEX) primers/probe and 60 ng of genomic DNA, to a final sample volume adjusted with nuclease-free water. This was loaded in to a DG8™ Cartridge with accompanying DG8™ Gasket and 70 μl of QX200™ Droplet Generation Oil for Evagreen for droplet generation using a QX200™ Droplet Generator. Ninety-six-well plates were then sealed using pierceable foil plate seals with a PX1™ PCR plate sealer. A T100™ Thermal Cycler was used with the following cycling conditions: enzyme activation for 5 min at 95 °C, followed by 40 cycles of denaturation at 94 °C for 30 s and annealing/extension at 55 °C for 1 min. Signal stabilization was achieved by cooling to 4 °C for 5 min, and heating to 98 °C for 5 min. A ramp rate of 2 °C per second was required for each step in the PCR. Data was then obtained using a QX200™ Droplet Reader with ddPCR™ Droplet Reader Oil and QuantaSoft™ Software, version 1.7.

**PCR and Sanger sequencing.** Polymerase chain reaction (PCR) was performed with 100 ng of template genomic DNA and gene specific primers using High-Fidelity PCR master mix (Qiagen) according to the manufacturer's recommendation. The list of gene variants tested along with the primer sequences are presented in S5 Table. The PCR products were analyzed in 2% agarose gel and purified using PCR purification kit (Qiagen). The purified products were subjected to Sanger sequencing and the results were analyzed using Chromas software (Technelysium Pty. Ltd., South Brisbane, Australia).

**Pathway analysis and selection of drug candidates.** The list of genes that carry missense, nonsense and splice site mutations and also mutated in at least two samples from the pairwise tumor and germline data of metastatic adrenocortical tumors was uploaded into the IPA software (Qiagen). The "core analysis" function included in the software was used to interpret the mutation data, which included the biological processes, canonical pathways, upstream transcriptional regulators, and gene networks. The drug candidates that are presented and can be targeted in at least two patients were only selected.

## Data availability

The datasets generated in this study is available in the dbGAP public repository with the accession ID: phs001658.v1.p1.

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

## Acknowledgements

This work was supported by the intramural research program of the Center for Cancer Research, National Cancer Institute, National Institutes of Health.

## Author contributions

Concept and design: S.K.G. and E.K. Development of methodology: S.K.G. and J.L. Generating and acquisition of data: S.K.G. Analysis and interpretation of data: S.K.G., J. L., and E.K. Manuscript writing and review: S.K.G., J.L., L.Z., E.H., M.C., and E.K. Study supervision: E.K.

## Additional information

**Competing interests:** The authors declare no competing interests.



