## [Peer Review File · Nature Communications]

Reviewers' comments:

Reviewer #1 (Remarks to the Author):

The manuscript from Gara et al describes the analysis of 33 metastatic adrenocortical cancers (ACC) from 14 patients to assess intratumour heterogeneity and how genomic profiles of metastatic tumours differ from those of primary tumours. Key conclusions are that the mutation load is 2.8 fold higher in metastatic than in primary tumour samples, that there is heterogeneity between metastases and that there are potential therapeutic targets that are not present in primary ACCs. Although they claim that this is the first study of the genomic landscape of metastatic ACCs, which is interesting, the paper currently has several major flaws which need to be addressed:

Major criticism:

The statement that the mutation load is 2.8 fold higher in metastatic than in primary tumours samples is a difficult one as the comparison is made between metastases in this cohort and primary tumours in the TCGA cohort. It could be that the cohort here also had a higher mutation load in the primary tumours which were not studied as far as I can tell. Could the authors analyse at least a few matched primary tumours from their cohort to assess if there is indeed such a dramatic difference?

How were transcribed and non-transcribed regions defined? This is not an easy task and from what I can see in this paper, the authors did not even have RNA expression data available.

How has the gene list in line 94 been selected. Are these oncogenes and tumour suppressor genes. Was there evidence of recurrent mutations in known or novel hotspots and were deleterious mutations observed in tumour suppressor genes? These are important points that need to be investigated as most of the observed mutations may just be passengers mutations without any relevance.

The heterogeneity plots in Figure 2 and 3 are very difficult to decipher and understand. I would recommend that the authors in the first instance show heatmaps or similar diagrams to demonstrate intratumour heterogeneity and potentially substantiate this with phylogenies.

The pathway analysis described on page 10 to identify novel drug candidates is very rudimentary. The genetic data entered does not seem to be pre-filtered for alterations that are likely to be functionally relevant. From this data, the authors then derive recommendations as to how these cancers can be more effectively treated. There is a very high probability of finding false positive associations with these pathway analysis approaches and this results would need to be validated at least in vitro before they are included into a publication.

Minor criticism:

The abstract mentions that tumour heterogeneity between metastases may explain resistance to systemic chemotherapy which is common in this cancer type. This statement is not supported by the data (ie the authors did not show any evidence that chemotherapy resistance drivers are common) and it is also not clear whether they refer to primary or acquired resistance here. It is more difficult to make a link between heterogeneity and primary resistance (in this case, I would rather expect divergent responses of different primary sites, but not necessarily a failure of all metastatic sites to respond) whereas it is conceivable that high heterogeneity could lead to rapid acquired resistance evolution (because of the higher probability of pre-existing drug resistant subclones). Such considerations can be elaborated in the discussion, but should not be added to the abstract.

I am quite surprised about the reported ratio of silent vs non-silent mutations of approx. 1:2. This

is exactly the opposite of what I would expect from exome sequencing where most variants are usually non-synonymous/non-silent. Can the authors explain? I would recommend randomly validating ~200 mutations to get clarity whether the mutation call algorithm is reliable. This should not be done by ddPCR as probe availability usually prefers validation of known drivers which are often real if detected but for example by sanger or amplicon sequencing.

The authors talk about fixation of mutations in MSH3 on page 6. I think this generates confusion as fixation usually refers to the expansion and eventually dominance of a mutation in a population. Fixation can never be achieved through LOH in an asexual species. What the authors describe is simply biallelic loss of an allele as far as I can tell.

Reviewer #2 (Remarks to the Author):

Interesting and important findings for a rare disease. The authors have effectively draw appropriate and possible clinical therapeutics into the WGS findings. This is important to bridge the discovered mutations into recommendations of care (the novel drug candidates). This work suggesting that metastatic tumors evolve and gain mutations could be reproduced with other tumors as the principle finding that metastatic sites differ from the primary lesion is of important oncologic relevance.

Can you comment on how these mutational pathways compare to other rare endocrine tumors? Is there a parallel?

The materials and methods section header appears to be misplaced.
Figure 5 may be thinned to only include A.

Reviewers' comments:

Reviewer #1 (Remarks to the Author):

The manuscript from Gara et al describes the analysis of 33 metastatic adrenocortical cancers (ACC) from 14 patients to assess intratumour heterogeneity and how genomic profiles of metastatic tumours differ from those of primary tumours. Key conclusions are that the mutation load is 2.8 fold higher in metastatic than in primary tumour samples, that there is heterogeneity between metastases and that there are potential therapeutic targets that are not present in primary ACCs. Although they claim that this is the first study of the genomic landscape of metastatic ACCs, which is interesting, the paper currently has several major flaws which need to be addressed:

We thank the reviewer for appreciating our work and also providing important suggestions to improve our manuscript. We have now addressed all the concerns raised by the reviewer and believe that our paper is now more clear and compelling.

Major criticism:

The statement that the mutation load is 2.8 fold higher in metastatic than in primary tumours samples is a difficult one as the comparison is made between metastases in this cohort and primary tumours in the TCGA cohort. It could be that the cohort here also had a higher mutation load in the primary tumours which were not studied as far as I can tell. Could the authors analyse at least a few matched primary tumours from their cohort to assess if there is indeed such a dramatic difference?

We thank the reviewer for raising this important point and we agree that it is more appropriate to check a few matched primary tumors in our cohort. However, we only have one subject's sample to perform this analysis given the rarity of the malignancy and the rarity of the clinical scenario in which patient would have both tumor sites removed to have tumor tissue available for analysis. Nonetheless, in the one subject, we have now performed whole-exome sequencing of the primary tumor and the lung metastatic tumor and found that overall mutation rate per megabase is 3-fold higher in metastatic lung tumor compared to its matched primary tumor which is consistent with our earlier findings where we observed a 2.8 fold higher mutation rate in metastatic ACC cohort when compared to TCGA primary ACC tumors. We have now included this result in Figure 1B and it is mentioned in the manuscript text (Please see page 5).

In addition, we compared the metastatic ACC mutation rate to other cancer types from TCGA including primary ACC. We believe that this analysis further put our findings into context that ACC metastasis relative not only to primary adrenocortical carcinoma but other primary cancer type has a higher mutation rate. We have now included this result in Figure 1C and it is mentioned in the manuscript text (Please see page 6).

We also performed an analysis of the mutation rate in the primary ACC cohort from the TCGA by tumor stage to test the idea that primary tumors with metastasis could in fact have a higher mutation burden as their metastasis. However, we found no significant difference in the mutation rate by tumor stage which further suggests that our findings are likely valid. We have now included this result in Figure 1D and it is mentioned in the manuscript text (Please see page 6).

Please note that the representative droplet digital PCR results in older version of Figure 1 was now moved to Supplementary Figure 1 due to the space issue and the order of the figures was corrected accordingly.

How were transcribed and non-transcribed regions defined? This is not an easy task and from what I can see in this paper, the authors did not even have RNA expression data available.

We thank the reviewer for raising this point. Yes, we do not have RNA expression data as we did not perform RNAseq in these tumors. As has been done by some investigators, we described transcribed and non-transcribed strand based on widely used annotation tools. In addition, the fact that libraries are stranded from the antisense strand, all the library effectively comes from (-) strand by which we calculated the mutations of genes on each strand. We have now included this in the methods section (Please see page 17)

How has the gene list in line 94 been selected. Are these oncogenes and tumour suppressor genes. Was there evidence of recurrent mutations in known or novel hotspots and were deleterious mutations observed in tumour suppressor genes? These are important points that need to be investigated as most of the observed mutations may just be passengers mutations without any relevance.

We thank the reviewer for making this important point. We selected genes (ENTHD1, HELZ2, PCDH12, SHANK1 and WDR66) with variants that were recurrent in multiple metastatic ACC tumors in different patients (Figure 1G) and importantly not present in primary ACC tumors from the adrenocortical cancer dataset of TCGA suggesting that these variants may be unique to metastatic tumors. Many of the variants we identified are also present in other

cancer types. For instance, P1772S variant in HELZ2 gene occur in cutaneous melanoma and colorectal adenocarcinoma tumors in the TCGA cohort. In addition, somatic mutations in ENTHD1 gene have also been identified in multiple breast cancer cases (Zhang et al., 2015), C1183S variant in PCDH12 was also present in renal cell carcinoma and colorectal carcinoma tumors. Although, the functional nature of these genes is not well-studied, some of them have been functionally implicated to have a role in cancer. For instance, WDR66 has been reported to be an oncogene and a novel marker for risk stratification in esophageal squamous cell carcinoma (Wang et al., 2013). We have now included this information in the discussion. (Please see page 13).

The heterogeneity plots in Figure 2 and 3 are very difficult to decipher and understand. I would recommend that the authors in the first instance show heatmaps or similar diagrams to demonstrate intratumour heterogeneity and potentially substantiate this with phylogenies.

We thank the reviewer for the suggestion. We have now replaced Figure 2B and 2D with an oncoplot and also added phylogenies. However, we found that Figures 2C, 2E and Figure 3 were informative when displayed in Venn diagrams as the number of tumor samples are 3 or less unlike in Figure 2B and 2D where the number of tumor samples are 6. In addition, since we hardly see any overlapping variants in Figure 2C and 2E, we did not find the oncoplot very informative with two samples.

The pathway analysis described on page 10 to identify novel drug candidates is very rudimentary. The genetic data entered does not seem to be pre-filtered for alterations that are likely to be functionally relevant. From this data, the authors then derive recommendations as to how these cancers can be more effectively treated. There is a very high probability of finding false positive associations with these pathway analysis approaches and this results would need to be validated at least in vitro before they are included into a publication.

We completely agree with the reviewer that subjecting genetic dataset to pathway analysis may have high probability of finding false positive associations. Therefore, we have strictly filtered out our genetic dataset prior to pathway analysis and selected only the list of genes that carry nonsynonymous damaging mutations for both pathway analysis and also for identifying drug candidates. However, the two tumors that belong to the same patient which exhibit hyper mutation phenotype, accounts for the large drug candidates. For that reason, in our downstream analysis, we have only selected the drugs that can be targeted in at least two patients for better clinical relevance. We have now clearly explained this in the methods section (Please see page 18). We believe that validating the proposed novel drug candidates in vitro and in vivo is beyond the scope of this study.

Minor criticism:

The abstract mentions that tumour heterogeneity between metastases may explain resistance to systemic chemotherapy which is common in this cancer type. This statement is not supported by the data (ie the authors did not show any evidence that chemotherapy resistance drivers are common) and it also not clear whether they refer to primary or acquired resistance here. It more difficult to make a link between heterogeneity and primary resistance (in this case, I would rather expect divergent responses of different primary sites, but not necessarily a failure of all metastatic sites to respond) whereas it is conceivable that high heterogeneity could lead to rapid acquired resistance evolution (because of the higher probability of pre-existing drug resistant subclones). Such considerations can be elaborated in the discussion, but should not be added to the abstract.

Yes, we agree with the reviewer and thank the reviewer for this excellent suggestion. However, we would like to note that we have only suggested that tumor heterogeneity between metastases in ACC may explain resistance to systemic chemotherapy and we have now included this in the discussion and removed it from the abstract as per the reviewer's suggestion (Please see pages 2 and 13-14).

I am quite surprised about the reported ratio of silen vs non-silent mutations of approx. 1:2. This is exactly the opposite of what I would expect from exome sequencing where most variants are usually non-synonymous/non-silent. Can the authors explain? I would recommend randomly validating ~200 mutations to get clarity whether the mutation call algorithm is reliable. This should not be done by ddPCR as probe availability usually prefers validation of known drivers which are often real if detected but for example by sanger or amplicon sequencing.

Although the accuracy for whole-exome sequencing is high, we randomly selected 77 variants with at least two variants from each metastatic ACC tumor to address the reviewer's concern. We have now tested 77 variants in 33 tumors that were detected by whole-exome sequencing independently by Sanger sequencing and were able to validate 72 of them. The list of genes and the primer sequences are now included in the Supplementary Table 5 and a total of 10 representative sanger sequencing results with chromatogram profile are presented in Supplementary Figure 2.

The authors talk about fixation of mutations in MSH3 on page 6. I think this generates confusion as fixation usually refers to the expansion and eventually dominance of a mutation in a population. Fixation can never be achieved through LOH in an asexual species. What the authors describe is simply biallelic loss of an allele as far as I can tell.

Yes, we agree with the reviewer and thank you for pointing this out. We have corrected this in the manuscript. (Please see page 7).

Reviewer #2 (Remarks to the Author):

Interesting and important findings for a rare disease. The authors have effectively draw appropriate and possible clinical therapeutics into the WGS findings. This is important to bridge the discovered mutations into recommendations of care (the novel drug candidates). This work suggesting that metastatic tumors evolve and gain mutations could be reproduced with other tumors as the principle finding that metastatic sites differ from the primary lesion is of important oncologic relevance.

We thank the reviewer for appreciating the work presented.

Can you comment on how these mutational pathways compare to other rare endocrine tumors? Is there a parallel?

*It is an excellent and very important question but more research needs to be done before we conclude on this aspect. Particularly in rare endocrine syndromes associated with adrenocortical tumors, it has been reported that inactivating mutations within the gene *PRKARIA* which is a regulatory subunit of Protein Kinase A are common in 45-80% of families with Carney complex. On the other hand, germ line inactivating mutations of the *TP53* tumor suppressor gene and loss of heterozygosity in *TP53* allele in the primary adrenocortical tumors are common in patients with Li-Fraumeni syndrome. Although we observed mutations in genes that are involved in protein kinase signaling, we did not see any mutations in the *TP53* gene in our metastatic tumors. Therefore, we believe the molecular pathways behind these rare endocrine syndromes do share some commonalities at the same time exhibit some specific differences for each tumor type and therefore there needs more studies to provide adequate comparison.*

The materials and methods section header appears to be misplaced.
Figure 5 may be thinned to only include A.

We thank the reviewer for pointing this out. We have now fixed the section header for the Methods section. We believe that Figure 5A displays a complex network of multiple pathways and provides a better overview while difficult to point out the important pathways. Therefore, we left Figure 5B-E for a comprehensive presentation of the pathways for the readership.

Reviewers' comments:

Reviewer #1 (Remarks to the Author):

The authors have tried to clarify my question about the definition of transcribed and non-transcribed regions. From their answer, it seems they actually mean the transcribed vs the non-transcribed strands of genes. However, this is still far from clear in the text (it sounds as if they are doing a comparison of genes which are transcribed in ACC vs those that are not transcribed). They also do not explain why such a comparison is important.

My question about potentially new driver mutations has only been partially addressed: They compared the identified mutations to data from other cancer types but did not assess if genes which have been suggested as tumour suppressors in the literature carry disruptive mutations such as frame shift-, nonsense- or splice site mutations in metastatic ACC. Whether mutations in known or suspected oncogenes were located at hotspots has also not been assessed.

The authors have provided more detail on the pathway enrichment analysis. They say that only 'damaging mutations' were included but they do not define what they mean by a damaging mutations or how they can be certain about a specific mutations having a damaging effect.

Overall, the pathway analysis results remain unconvincing.

Line 100: I think this should say "genes recurrently mutated" instead of "genes with recurrent mutations"

Figure 1 B: I think the labelling must be wrong: >200 mutations/Mb seems extremely unlikely and does not fit with data shown in Fig 1A.

Figure 2:

The authors have added heatmaps but these and the Venn diagrams are poorly labelled and described (ie are these non-silent mutations only or all mutations?). Also, there seems to be quite a lot of overlap between samples from different cases in 2B and 2D but this is neither commented on nor is it clear which genes this affects. The new heatmaps that have now been included suggest that a large part of this overlap may be due to the inclusion of the hypermutated case – the analysis should have also been performed without this case.

Figure 3:

The authors did not include heatmaps and phylogenies for the cases in Fig 3. This appeared unnecessary for the authors as five cases only had two samples, but these diagrams, appropriately labelled with gene names, would have given the opportunity to understand at what point specific genes acquired a mutation (ie on the trunk or on branches). Moreover, the lack of a heatmap and phylogeny make it difficult to judge what may explain why 16 mutations are shared by various pairs of two out of the three samples in Fig 3B. This indicates phylogenetic conflicts which can either indicate poor tumour content of the analysed samples or copy number aberration heterogeneity that leads to mutations loss. A phylogenetic reconstruction attempt could have provided the necessary clarity here.

Reviewer #2 (Remarks to the Author):

Gara et al have revised the original manuscript titled Metastatic ACC displays higher mutation rate and tumor heterogeneity than primary tumors. The work remains an important bridge to identify novel drug candidates for this fatal disease. The work is of oncologic relevance. This work represents a starting point for others to evaluate primary and metastatic adrenal carcinoma with regard to inactivation mutations and molecular pathways that can be targeted. Matched cohorts will be necessary for confirmation of these findings. The authors have acknowledged that in the revision. Gara et al have chosen a title that appears to compare primary and metastatic tumor from the same patient but actually compares their metastatic tumor cohort to the TCGA dataset. Perhaps they will consider adjusting the title to clarify this point.

Reviewers' comments:

Reviewer #1 (Remarks to the Author):

The authors have tried to clarify my question about the definition of transcribed and non-transcribed regions. From their answer, it seems they actually mean the transcribed vs the non-transcribed strands of genes. However, this is still far from clear in the text (it sounds as if they are doing a comparison of genes which are transcribed in ACC vs those that are not transcribed). They also do not explain why such a comparison is important.

We thank the reviewer for the comment and we apologize that it was not clearly explained in the manuscript text. We analyzed the transcribed vs the non-transcribed strands of genes. We have now modified the manuscript text accordingly (Please see Page 6, lines 95-99). We performed such an analysis to understand the relevance of metastatic ACC tumors to primary ACC tumors and also other solid tumors (Please see page 6, line 99). Our analysis showed that similar to primary ACC and most other solid tumors, C>T transitions are more common in metastatic ACC tumors.

My question about potentially new driver mutations has only been partially addressed: They compared the identified mutations to data from other cancer types but did not assess if genes which have been suggested as tumor suppressors in the literature carry disruptive mutations such as frame shift-, nonsense- or splice site mutations in metastatic ACC. Whether mutations in known or suspected oncogenes were located at hotspots has also not been assessed.

We thank the reviewer for the comment and we hope we now have completely addressed the issue raised. We found genes such as CSMD2 (CUB and Sushi Multiple Domains 2), LRP1b (LDL Receptor related Protein 1B) and KIAA0100 which have been suggested to have tumor suppressor functions in the literature and that were also found to be mutated in metastatic ACC tumors. Particularly, CSMD2, a candidate tumor suppressor gene in colorectal cancer patients was mutated in three metastatic ACC patients (Figure 1G). LRP1b, a candidate tumor suppressor gene that is frequently inactivated in non-small lung cancer cells was also mutated in three metastatic ACC patients (Figure 1G). Finally, KIAA0100, another candidate tumor suppressor gene in acute monocytic leukemia was found to be mutated in three metastatic ACC patients (Figure 1G). However, all of these mutations in metastatic ACC are MISSENSE (Please see page 12-13, lines 251-262). We did not find any known or suspected oncogenes in the literature to be frequently mutated in metastatic ACC. On the other hand, a review of literature in ACC suggests CTNNB1, TP53 and ZNRF3 as the three major known tumor suppressors genes that are frequently mutated in ACC (Please see page 13, lines 262-263). However, it must be noted that the literature until now focuses on primary ACC tumors but not on metastatic ACC. Nevertheless, we observed multiple known CTNNB1 mutations (p.D32G, p.G34R, p.S45P and p.S45F), one of the most frequently mutated gene in our metastatic cohort, which is evident in Figure 1G. We did not find any known or novel mutations in TP53 and ZNRF3 genes (Please see page 13, lines 263-266). The majority of the recurrent mutations were missense mutations. Although, we observed frame shift-, nonsense- or splice site mutations in our metastatic cohort in different genes, the majority were

not recurrent (present in multiple metastatic tumors) and are therefore not discussed in this manuscript. We have now modified the manuscript text accordingly (Please see page 13, line 269-273) .

The authors have provided more detail on the pathway enrichment analysis. They say that only ‘damaging mutations’ were included but they do not define what they mean by a damaging mutations or how they can be certain about a specific mutations having a damaging effect. Overall, the pathway analysis results remain unconvincing.

We thank the reviewer for pointing this out. We defined “damaging mutations” as any missense, nonsense mutation that occur within the exonic region of any gene and also splice site mutations that occur in the exon-intron boundaries within a gene. We agree with the reviewer that one cannot absolutely be certain about a specific mutation to have a damaging effect until relevant functional studies are performed. Therefore, we have now revised the manuscript text containing “damaging mutations”, to be more specific, to “missense, nonsense mutations that occur within the exonic region of any gene and also splice site mutations that occur in the exon-intron boundaries within a gene” (Please see page 19, lines 411-412).

Although, pathway analysis is done with gene expression data, multiple studies have used whole exome sequencing and mutation data to understand the relevance of overall mutation signatures to molecular pathways (Please see page 15, lines 303-305). We have also cited some relevant studies.

Line 100: I think this should say “genes recurrently mutated” instead of “genes with recurrent mutations”

We thank the reviewer for pointing this out. We have now corrected this in the manuscript (Please see page 6, line 103 in this revised version)

Figure 1 B: I think the labelling must be wrong: >200 mutations/Mb seems extremely unlikely and does not fit with data shown in Fig 1A.

We thank the reviewer for pointing this out. We have now corrected this in the revised version (Please see Figure 1B).

Figure 2: The authors have added heatmaps but these and the Venn diagrams are poorly labelled and described (ie are these non-silent mutations only or all mutations?). Also, there seems to be quite a lot of overlap between samples from different cases in 2B and 2D but this is neither commented on nor is it clear which genes this affects. The new heatmaps that have now been included suggest that a large part of this overlap may be due to the inclusion of the hypermutated case – the analysis should have also been performed without this case.

We thank the reviewer for pointing this. We have now replaced the Venn diagrams in Figure 2 with heatmaps and phylogenies and we have labelled and described the heatmaps in the figure legend and manuscript text respectively (Please see page 26, lines 599-605 and Figure 2). In addition, we completely agree with the reviewer that there are few overlapping genes in Figure 2B and 2D and the large part of this overlap is due to the inclusion of the hypermutated case. We thank the reviewer for

the excellent suggestion to perform the analysis without the hypermutated case. We have now performed this analysis and zoomed in the distribution of overlapping genes (Please see page 8, lines 152-155). For metastatic lung ACC tumors, WDR66 is the only gene that is mutated in at least three cases when the hypermutated case was excluded (Figure 2B). All the other genes that are mutated appear in only two cases. Similarly, all the shared genes that are mutated in metastatic ACC other tumor sites appear only in two cases (Figure 2C). CTNNB1 is the only gene that is mutated in metastatic ACC liver in both cases (Figure 2D) whereas there are no shared mutations in genes in metastatic ACC peritoneum (Figure 2E). Please note that we have now moved the heatmaps containing the hypermutated case to Supplemental Figure 7 due to space constraints and Figure 2D (old version) is now Figure 2C and Figure 2C (old version) is now Figure 2D.

Figure 3: The authors did not include heatmaps and phylogenies for the cases in Fig 3. This appeared unnecessary for the authors as five cases only had two samples, but these diagrams, appropriately labelled with gene names, would have given the opportunity to understand at what point specific genes acquired a mutation (i.e. on the trunk or and branches). Moreover, the lack of a heatmap and phylogeny make it difficult to judge what may explain why 16 mutations are shared by various pairs of two out of the three samples in Fig 3B. This indicates phylogenetic conflicts which can either indicate poor tumor content of the analysed samples or copy number aberration heterogeneity that leads to mutations loss. A phylogenetic reconstruction attempt could have provided the necessary clarity here.

We thank the reviewer for raising this point. We have now replaced all the Venn diagrams in Figure 3 with heatmaps and phylogenies whenever there are more than two samples. Among the 16 mutations that are shared between the three tissues, genes such as CTNNB1, IGF2R and SF1 that are known to play an important role in the pathogenesis of adrenocortical tumor were present (Figure 3A). Higher number of additional shared mutations in genes (five) between kidney and liver compared to only one mutation in gene between kidney and peritoneum (Figure 3A). Although there are 20 shared mutations in genes between lung and other tissue site, CTNNB1 is the only known common gene that is mutated between these tumor tissues (Figure 3B; Please see page 9, lines 169-176)). Similarly, the lung and other tissue site from the patient with hypermutated phenotype shared 279 mutations in genes between each other (Figure 3D). The tumor content of the analyzed samples was more than 90% which was confirmed by an endocrine pathologists as described in the methods section, page 17, lines 348-350.

Reviewer #2 (Remarks to the Author):

Gara et al have revised the original manuscript titled Metastatic ACC displays higher mutation rate and tumor heterogeneity than primary tumors. The work remains an important bridge to identify novel drug candidates for this fatal disease. The work is of oncologic relevance. This work represents a starting point for others to evaluate primary and metastatic adrenal carcinoma with regard to inactivation mutations and molecular pathways that can be targeted. Matched cohorts will be necessary for confirmation of these findings. The authors have acknowledged that in the revision. Gara et al have chosen a title that appears to compare primary and metastatic tumor from the same patient but actually compares their metastatic tumor cohort to the TCGA dataset. Perhaps they will consider adjusting the title to clarify this point.

We thank the reviewer for appreciating and particularly emphasizing the significance and clinical relevance of our work. However, we believe that the current title justifies our findings since the matched primary and metastatic tumor from our cohort that we have performed during the revision have corroborated with the results when we compared our metastatic tumors with primary tumors of the TCGA dataset. Therefore, we decided to keep the title as it is and we hope the reviewer will agree with our point.

REVIEWERS' COMMENTS:

Reviewer #1 (Remarks to the Author):

The authors have now answered all my questions and have updated the manuscript accordingly.